

# Renormalization group for open quantum systems using environment temperature as flow parameter

Konstantin Nestmann[1,2] and Maarten R. Wegewijs[1,2,3]

**1** Institute for Theory of Statistical Physics, RWTH Aachen, 52056 Aachen, Germany
**2** JARA-FIT, 52056 Aachen, Germany
**3** Peter Grünberg Institut, Forschungszentrum Jülich, 52425 Jülich, Germany

## Abstract

We present the $T$-flow renormalization group method, which computes the memory kernel for the density-operator evolution of an open quantum system by lowering the physical temperature $T$ of its environment. This has the key advantage that it can be formulated directly in real time, making it particularly suitable for transient dynamics, while automatically accumulating the full temperature dependence of transport quantities. We solve the $T$-flow equations numerically for the example of the single impurity Anderson model. We benchmark in the stationary limit, readily accessible in real-time for voltages on the order of the coupling or larger using results obtained by the functional renormalization group, density-matrix renormalization group and the quantum Monte Carlo method. Here we find quantitative agreement even in the worst case of strong interactions and low temperatures, indicating the reliability of the method. For transient charge currents we find good agreement with results obtained by the 2PI Green's function approach. Furthermore, we analytically show that the short-time dynamics of both local and non-local observables follow a "universal" temperature-independent behaviour when the metallic reservoirs have a flat wide band.


## Contents



# 1 Introduction

The physics of open quantum systems is crucial to many fields and much progress has been made in the description of their transient and non-equilibrium dynamics. By now certain situations are well understood and characterized, such as the important case of memoryless Markovian semigroup dynamics governed by the Gorini-Kossakowski-Sudarshan-Lindblad (GKSL) [1, 2] quantum master equation (QME), which typically arises in the limits of weak coupling or high temperature. As the temperature is lowered towards the system-reservoir coupling scale, the dynamics is no longer well described by a semigroup. The required non-semigroup corrections are however often unproblematic to compute using well-developed perturbation expansions of either the memory kernel [3] or the time-local generator [4], whose precise connection has only recently been worked out [5, 6]. In contrast, the description of physics at low temperatures, strong coupling and large interactions remains challenging although much progress has been made. Various advanced methods focusing on either stationary or time-dependent quantities have been devised, such as the functional [7–9], numerical [10, 11] or density matrix renormalization groups [12, 13], quantum Monte Carlo methods [14] or path integral approaches [15, 16], to name but a few. Here we focus in particular on semi-analytical calculations of the dynamics of the density operator, in particular the real-time renormalization group (RTRG) method [3] based on a diagrammatic expansion of the memory kernel. Its most recent formulation, the so-called $E$-flow RG scheme [17], has been successfully applied to a range of models [18–21] and provides a number of key technical ideas for the present paper.

However, already an earlier version of the RTRG approach led to the interesting insight [22, 23] that the Liouville density-operator space for a large class of fermionic models can be elegantly generated using "superfermionic" excitations [24], simplifying this theory both in frequency *and* time representation. This led on the one hand to the discovery of the non-perturbative fermionic duality [25, 26], an exact "dissipative symmetry" which cross-relates different eigenvalues and -vectors of the memory kernel and propagator in a simple fashion. Corresponding exact relations for the Kraus measurement operators of the dynamics, the time-local generator and its Lindblad jump rates and -operators have recently been worked out [27]. On the other hand, it was found that approximate calculations could be improved beyond the standard bare perturbation theory [28, 29]: by directly going to the wideband limit a much

simplified *renormalized* perturbation series was obtained [6, 22, 23, 30]. Unlike the bare expansion around the decoupled limit $\Gamma \to 0$ of reversible dynamics, here one expands around the *high-temperature limit* $T \to \infty$, which already includes dissipative behaviour into the reference solution of the series. This leads to completely different – and often better behaved – memory-kernel approximations [6, 22, 23, 30] and facilitates solution of exactly solvable cases [23, 27, 30].

In the present paper we pursue this development further and introduce the $T$-flow method. It allows to compute the memory kernel using the physical temperature $T$ as a renormalization-group flow parameter starting from $T = \infty$, where the evolution is known to be a GKSL-semigroup [22, 23] with simple jump operators [27, 30]. The basic idea is to obtain the low-temperature dynamics by literally lowering the physical temperature of the system's environment in small steps and systematically computing the memory kernel corrections that this generates. For this task the above mentioned renormalized perturbation expansion forms the natural starting point and we show how it can be combined with key techniques of the later developed $E$-flow scheme: by simply taking a $T$ derivative (instead of an $E$ derivative) of the full diagrammatic series for the memory kernel and the effective vertices one readily obtains a self-consistent hierarchy of differential equations, which can be systematically approximated while self-consistently keeping full time-propagators between the effective vertices.

In contrast to the $E$-flow method, we can formulate everything directly either in the time or frequency domain. Here we focus on numerically solving the $T$-flow RG equations in time space for the example of an interacting Anderson dot. Notably, in doing so we automatically generate solutions for all temperatures. One thus works directly in terms of the temperature dependence of relevant time-evolving quantities, which are closely connected to the many-body physics of interest and their experimental signatures. Clearly, this built-in feature of the $T$-flow is of special interest for thermoelectric calculations which are, however, beyond the scope of the present work.

The paper is organized as follows. In Sec. 2 we connect the time correlations of the environment to the key idea of the $T$-flow. Here we merely try to provide some physical intuition for the later technical developments. In Sec. 3 we introduce the exemplary Anderson model, its superfermion Liouville-space formulation and the renormalized perturbation theory around $T = \infty$. From this natural starting point we derive the $T$-flow equations in Sec. 4 borrowing techniques from the $E$-flow. We present our first results in Sec. 5, focusing on charge currents, occupations and charge fluctuations after verifying the legitimacy of the computed propagators (complete positivity). We investigate the reliability of our approach by comparison with various other methods and as a first application we investigate the impact of the interaction on the phenomenon of reentrant charge decay [30]. We summarize and point out future directions in Sec. 6 and discuss relations of our technical results to the broader understanding of memory effects in open quantum systems [31]. We set $\hbar = k_B = 1$.

## 2 Temperature as flow parameter: Time-correlations

We first consider for simplicity a system in contact with a single environment, the latter initially in equilibrium at temperature $T$. As mentioned in the introduction, the basic idea of the $T$-flow is to calculate low-temperature dynamics by literally lowering the temperature of the environment step-by-step. To develop some intuition for this we focus on the environment correlations. This suggests that our method is similar in spirit to Wilson's RG for critical systems [32] with the key difference that correlations in time – instead of space – are at the focus. In particular, the temperature $T$ sets the inverse correlation time and we will show in the next section that these time-correlations are effectively encoded into a single temperature-

dependent correlation function $\gamma^-(t,T)$. The crucial underlying assumption for this is that the reservoirs are *non-interacting*: in this case all multi-particle correlation functions of the reservoirs – determining the time-nonlocal backaction of the system via the memory kernel $\Sigma$ – factorize into one-particle functions (Wick-theorem).

Following Wilson's idea we set up a flow from the high-temperature limit where the correlations are short-ranged ($\gamma^- = 0$ for $T \to \infty$) to one with long-ranged power laws ($\gamma^- \propto 1/t$ for $T \to 0$). Thus, in the low-temperature regime of interest different time scales contribute equally when integrating power law correlations (e.g. $\int_t^{10t} dt'/t' = \ln 10$ independent of $t$) to compute the dynamics. An important difference with Wilson's RG is that we do not introduce an artificial flow parameter into the description, but instead use a variable that is already part of the problem. This is similar in spirit to the $E$-flow RG method for open quantum systems [17], which by choosing the Laplace variable $E$ as flow parameter is intrinsically bound to the frequency domain, in the sense that only at the end of the calculation one can go to the conjugate time-domain. By instead choosing the physical environment temperature $T$ we remain flexible to work in either domain.

Whereas at high temperatures the dynamics can be computed via a memory kernel $\Sigma$ using perturbation theory, this becomes unreliable at low temperatures, because the slowly decaying correlations amplify higher order contributions requiring a more systematic treatment. In the $T$-flow this is done by integrating out thermal fluctuations in many small steps $\delta T$, as opposed to treating the entire correction $T = \infty \to 0$ in one piece. Thus the reduction of thermal fluctuations generates effective higher-order coupling effects. These corrections are controlled by the temperature sensitivity of the correlation function, $\partial_T \gamma^-$. A key observation is that this quantity never behaves as a power law in time and even vanishes at $T = 0$ for all times, in contrast to $\gamma^-$ itself which is divergent for $t \to 0$ and slowly decaying for $t \to \infty$. In this sense, the computation of the temperature sensitivity of the memory kernel, $\partial_T \Sigma$, is better behaved than that of $\Sigma$ itself. This is very similar to the calculation the memory kernel in the $E$-flow method [17].

At this point it is important to note that we do not change temperature as function of time. Instead we consider the entire dynamics – via its memory-kernel – at temperatures $T$ and $T - \delta T$ when we stepwise lower the temperature. Proceeding this way it is by no means obvious that we do not pass through $T = 0$ and continue to negative temperatures. However, the above mentioned properties of the temperature sensitivity of the correlation function ensure that this does not happen: as we will see, the $T$-flow terminates at the fixed point $T = 0$:

$$\lim_{T \to 0} \frac{\partial \Sigma}{\partial T}(t, T) = 0. \tag{1}$$

Finally, before turning to the technical implementation, we highlight that because in our $T$-flow method temperature itself serves as a flow parameter it does not play the role of a cut off in the technical RG sense (there is no other running energy scale). This should not be confused with the fact explained above that temperature sets the inverse correlation time beyond which time integrations stop contributing, i.e., it does cut off time integrations in the ordinary non-RG sense (time is not a flow parameter).

## 3 Renormalized perturbation theory around $T = \infty$

With these ideas in mind, we now concretely set up the approach for the Anderson impurity model, noting that the following analysis can be extended straightforwardly to a large class of relevant fermionic transport models, see Refs. [23, 25, 27] for details. The system consists of

a single orbital quantum dot with energy $\epsilon$ and Coulomb repulsion $U$ described by

$$H = \epsilon(n_\uparrow + n_\downarrow) + Un_\uparrow n_\downarrow, \tag{2}$$

where $n_\sigma = d_\sigma^\dagger d_\sigma$ denotes the number operator for spin $\sigma$. Several non-interacting electron reservoirs

$$H_R = \sum_{r\sigma} \int d\omega(\omega + \mu_r)a_{r\sigma}^\dagger(\omega)a_{r\sigma}(\omega), \tag{3}$$

are connected to the dot via tunnel junctions

$$H_T = \sum_{r\sigma} \int d\omega \sqrt{\frac{\Gamma_{r\sigma}}{2\pi}} \left(a_{r\sigma}^\dagger(\omega)d_\sigma + d_\sigma^\dagger a_{r\sigma}(\omega)\right). \tag{4}$$

We take the reservoirs labelled by $r$ to be initially in grand canonical thermal equilibrium at different temperatures $T_r$ and chemical potentials $\mu_r$. The spin-dependent spectral densities $\Gamma_{r\sigma}$ are assumed to be energy independent (wideband limit) and real-valued. The total system is described by

$$H_{\text{tot}} = H + H_R + H_T. \tag{5}$$

Assuming that the total initial state is a product state, $\rho_{\text{tot}}(0) = \rho_0 \otimes \rho_R$, there is a well-defined propagator $\Pi(t)$ of the system density operator, $\rho(t) = \Pi(t)\rho_0$ for arbitrary initial states $\rho_0$. The propagator satisfies the time-nonlocal quantum master equation [3, 33, 34]

$$\dot{\Pi}(t) = -iL\Pi(t) - i\int_0^t ds\mathcal{K}(t,s)\Pi(s), \tag{6}$$

where $L\bullet := [H, \bullet]$ is the local Liouvillian and $\mathcal{K}$ denotes the memory kernel. In our case the memory kernel depends only on time differences, $\mathcal{K}(t,s) \equiv \mathcal{K}(t-s)$, since the system is time-translation invariant. By directly computing the propagator, we can analyse the time-evolution for arbitrary initial states $\rho_0$.

A standard way to compute $\mathcal{K}$ is via a bare perturbation theory in the coupling $\Gamma$ [28, 35]. We will use this as an example to introduce the superfermion description of Liouville space and some notation. With the fermion parity operator $(-\mathbb{1})^n := (\mathbb{1} - 2n_\uparrow)(\mathbb{1} - 2n_\downarrow)$ and fermion operators

$$d_{\eta\sigma} := \begin{cases} d_\sigma^\dagger & \text{for} \quad \eta = + \\ d_\sigma & \text{for} \quad \eta = - \end{cases}, \tag{7}$$

superfermions denote superoperators acting on an operator argument $\bullet$ as

$$G_{\eta\sigma}^p \bullet := \frac{1}{\sqrt{2}}\left[d_{\eta\sigma} \bullet + p(-\mathbb{1})^n \bullet (-\mathbb{1})^n d_{\eta\sigma}\right]. \tag{8}$$

Roughly speaking, $p = +$ gives a creation and $p = -$ an annihilation superoperator, similarly to their operator analogues (7). For example, superfermions anticommute as

$$\left\{G_1^{p_1}, G_2^{p_2}\right\} = \delta_{p_1\bar{p}_2}\delta_{1\bar{2}}, \tag{9}$$

where we use multi-indices $1 \equiv (\eta_1, \sigma_1)$ and the notation $\bar{p} := -p$, $\bar{1} := (-\eta_1, \sigma_1)$. They furthermore respect a super-Pauli principle, $(G_1^p)^2 = 0$, which formally means that it is impossible to create (or annihilate) the same superfermion twice [23]. Any superoperator can

be expressed in terms of strings of creation or annihilation superfermions. For example, the Liouvillian $L$ can be written as

$$L = \sum_{\eta\sigma} \left[ \bar{\eta} \left( \epsilon + \tfrac{1}{2} U \right) G_{\bar{\eta}\sigma}^+ G_{\eta\sigma}^- + \tfrac{1}{2} U \left( G_{\bar{\eta}\sigma}^+ G_{\eta\sigma}^- G_{\bar{\eta}\bar{\sigma}}^- G_{\eta\bar{\sigma}}^- + G_{\bar{\eta}\bar{\sigma}}^+ G_{\eta\sigma}^+ G_{\bar{\eta}\sigma}^+ G_{\eta\sigma}^- \right) \right].$$

The memory kernel can be computed via its diagrammatic representation as the sum over connected diagrams, which is a series around the uncoupled limit:

$$-i\mathcal{K} = \boxed{\phantom{xx}} + \boxed{\phantom{xxxxx}} + \boxed{\phantom{xxxxx}} + \cdots. \tag{10}$$

Here the single horizontal line, directed from right to left (not indicated), represents the reference dynamics $\Pi_0(t) := e^{-iLt}$, see Refs. [3, 23] for details. For example, using the superfermions the lowest order diagram is explicitly given by

$$-i\mathcal{K}^{(1)} = \boxed{\phantom{xx}} = -\gamma_1^p(t) G_1^+ \Pi_0(t) G_{\bar{1}}^{\bar{p}}, \tag{11}$$

where we implicitly sum over all repeated (multi-)indices. The two possible contraction functions are given by $\gamma_1^p(t) \equiv \gamma_{\eta\sigma}^p(t) = \sum_r \gamma_{\eta\sigma r}^p(t)$ with

$$\gamma_{\eta\sigma r}^+(t) = \tfrac{1}{2} \Gamma_{r\sigma} \bar{\delta}(t), \qquad (p = +) \tag{12a}$$

$$\gamma_{\eta\sigma r}^-(t) = -i \frac{\Gamma_{r\sigma} T_r}{\sinh(\pi t T_r)} e^{i\bar{\eta}\mu_r t}. \quad (p = -) \tag{12b}$$

These are essentially the retarded and Keldysh reservoir correlation functions [22, 30], respectively. The $\bar{\delta}$ distribution – defined such that $\int_0^t \bar{\delta}(t-s) f(s) ds = f(t)$ – occurs in the *temperature independent* $\gamma_{\eta\sigma}^+$ contraction and is a consequence of the wideband limit, which was taken from the very beginning in the definition of the Hamiltonian in Eq. (4). It leads to a separate time-local contribution present in the memory kernel:

$$\mathcal{K}(t-s) = (L + \Sigma_\infty) \bar{\delta}(t-s) + \Sigma(t-s), \tag{13}$$

where

$$-i\Sigma_\infty := -\frac{1}{2} \sum_r \Gamma_{r\sigma_1} G_1^+ G_{\bar{1}}^-. \tag{14}$$

Adding this to the Liouvillian of the uncoupled system we obtain the new reference Liouvillian

$$L_\infty := L + \Sigma_\infty, \tag{15}$$

which generates the exact GKSL-semigroup dynamics of the model at infinite temperature [23],

$$\Pi_\infty(t) := \lim_{T_r \to \infty} \Pi(t) = e^{-iL_\infty t}. \tag{16}$$

In other words, by the simple renormalization (15) of the kinetic equation (6) we obtain the exact result at $T = \infty$ for the propagator $\Pi(t)$. Interestingly, it is now possible to formulate a *renormalized* perturbation theory around the infinite temperature limit [3, 23] by resumming all the time-local $\gamma_{\eta\sigma}^+$ contractions exactly. To do so the diagrammatic rules have to be changed as follows: first, only $\gamma_{\eta\sigma}^-$ contractions and superfermionic creation operators $G_{\eta\sigma}^+$ are allowed. Second, all of the intermediate uncoupled propagators $\Pi_0$ are replaced by infinite temperature propagators $\Pi_\infty$. Thus, the renormalized version of Eq. (11) reads

$$-i\Sigma^{(1)}(t) = \boxed{\phantom{xx}} = -\gamma_1^-(t) G_1^+ \Pi_\infty(t) G_{\bar{1}}^+. \tag{17}$$

Compared to the bare perturbation theory this already incorporates dissipation into the reference solution, often[1] improving the quality of the approximation. For example, a finite number of terms of the renormalized series gives the exact solution in three different physical limits: By construction it is exact in the limits of vanishing coupling $\Gamma \to 0$ or infinite temperature $T \to \infty$, but it can be shown that it is additionally exact in the non-interacting limit $U \to 0$ [23, 30] for any $\Gamma$ and $T$, which is not the case in any finite order bare perturbation theory in $\Gamma$.

Whereas the memory kernel can be used to compute the density operator and thus expectation values of local observables, it is not sufficient to determine expectation values of nonlocal observables such as transport quantities. For these additional observable-kernels are needed [3]. Here we will focus on the particle current flowing out of reservoir $r$ defined by $I_r(t) := -\partial_t \langle N_r \rangle_{\mathrm{tot}}(t)$, which can be obtained using a current kernel $\mathcal{K}_{I_r}$ via

$$I_r(t) = -i \operatorname{Tr} \int_0^t ds \, \mathcal{K}_{I_r}(t-s)\rho(s). \tag{18}$$

Similarly to the the ordinary memory kernel $\mathcal{K}$ [cf. Eq. (13)] we can decompose

$$\mathcal{K}_{I_r}(t-s) = \Sigma_{I_r,\infty}\bar{\delta}(t-s) + \Sigma_{I_r}(t-s). \tag{19}$$

Here the first term is time-local due to the wideband limit and gives the infinite temperature part of to the current-kernel. This term corresponds to the renormalization (15) of the kinetic equation (6) that we performed to obtain the exact result at $T = \infty$. When it is applied to a finite-$T$ state, we obtain a contribution to the current expectation value that probes the deviation of the spin-orbital occupations from half filling,

$$I_{r,\infty}(t) := -i \operatorname{Tr} \Sigma_{I_r,\infty}\rho(t) \tag{20}$$

$$= -\frac{1}{4}\eta_1\Gamma_{r\sigma_1} \operatorname{Tr} G_1^- G_{\bar{1}}^- \rho(t) \tag{21}$$

$$= \sum_\sigma \Gamma_{r\sigma}\left(\frac{1}{2} - \langle n_\sigma\rangle(t)\right). \tag{22}$$

Note that $\lim_{t\to 0^+} I_r(t) = I_{r,\infty}(0) \neq 0$ in general: the current instantly rises at $t = 0$ (no coupling) to a finite value because we are working in the wideband limit. For large but finite bandwidth $D$ the current approaches our result on the very short timescale $1/D$ [23, 36, 37].

The time-nonlocal current-kernel $\Sigma_{I_r}$ can be computed using the same diagrammatic series that is used for $\Sigma$, except that the leftmost vertex and its contraction need to be replaced [3]. In the superfermionic notation we use here this amounts to replacing the leftmost $G_{\eta\sigma}^+ \to \frac{\eta}{2}G_{\eta\sigma}^-$ and $\gamma_{\eta\sigma}^- \to \gamma_{\eta\sigma r}^-$. Thus the leading order renormalized diagram of the current kernel reads

$$-i\Sigma_{I_r}^{(1)}(t) = \;\underset{\times\;\longrightarrow\;\bullet}{\boxed{\phantom{xxx}}}\; = -\gamma_{\eta_1\sigma_1 r}^-(t)\frac{\eta_1}{2}G_1^-\Pi_\infty(t)G_{\bar{1}}^+, \tag{23}$$

where we use a cross to indicate the replaced vertex.

# 4 The $T$-flow renormalization group

We are now ready to derive the self-consistent $T$-flow equations for $\partial_T\Sigma$, which allows us to lower the temperature in small steps $\delta T$, schematically via

---

[1]This is true for the renormalized *time-nonlocal* memory kernel approach, but *not* for the renormalized version of *time-local* approach (TCL), see Ref. [6].

$\Sigma(t, T - \delta T) = \Sigma(t, T) - \partial_T \Sigma(t, T) \delta T$. This is inspired by the derivation of the $E$-flow method [17], in particular by the definition of the effective vertices and the usage of full propagators between them. For simplicity we consider the case where all reservoirs have a common temperature $T_r = T$ – the general case is explained in App. D – while allowing for arbitrary applied bias $V = \mu_L - \mu_R$. We are thus considering transient dynamics to a non-equilibrium stationary state.

We first bring the renormalized series in self-consistent form by resumming all connected subblocks without uncontracted lines, thereby replacing infinite temperature propagators $\Pi_\infty$ by full ones represented by double lines, $\Pi \equiv \text{====}$. We then have

$$-i\Sigma = \boxed{\phantom{xx}} + \boxed{\phantom{xxxxx}} + \cdots. \tag{24}$$

Thus for example, the third term of Eq. (10) is already contained within the first term of Eq. (24) and so on. Next, we introduce effective $n$-point supervertices $G_{1...n}$ as sums over all connected diagrams with $n$ uncontracted lines. Specifically we will need

$$G_1 \equiv \phantom{x} := \phantom{x} + \boxed{\phantom{xxx}} + \boxed{\phantom{xxxxx}} + \cdots, \tag{25}$$

$$G_{12} \equiv \phantom{x} := \boxed{\phantom{xxxx}} + \cdots. \tag{26}$$

Note that the $T$-dependent effective supervertex $G_1$ (without superscript) differs from the $T$-independent superfermion $G_1^+$ (the first term in (25), defined by Eq. (8) with $p = +$) by finite-temperature corrections. Some remarks are necessary to make the above definitions more precise and these are given in App. A. Importantly, one can express $G_1$ using $G_{12}$ and $\Pi$ in a self-consistent manner as

$$\phantom{x} = \phantom{x} + \boxed{\phantom{xxx}} + \boxed{\phantom{xxx}} + \boxed{\phantom{xxx}}. \tag{27}$$

This can be seen in the following way: cutting off the leftmost vertex in each term of Eq. (25) from the rest of the diagram (except in the trivial first term), the remaining part on the right will have two uncontracted lines and will either be disconnected or it will remain connected. In the former case the diagram before cutting will be included in the second term in Eq. (27), whereas in the latter case it must belong either to the third or fourth term. By this way of sorting, all diagrams are included without double counting. Combining Eqs. (24)–(25), we see that the memory and current kernel can be expressed using effective supervertices as

$$-i\Sigma = \boxed{\phantom{xxx}}, \quad -i\Sigma_{I_r} = \boxed{\phantom{xxx}}. \tag{28}$$

Note that the resummation to full propagators performed to obtain Eq. (24) is crucial for Eqs. (28) to hold.

In the following we will focus on the memory kernel, noting that the treatment for the current-kernel is formally very similar. Taking a $T$ derivative of Eq. (28), which we diagrammatically represent using a slashed line, we obtain the key relation

$$-i\frac{\partial \Sigma}{\partial T} = \boxed{\phantom{xxx}} + \boxed{\phantom{xxx}} + \boxed{\phantom{xxx}}. \tag{29}$$

The first term contains the temperature derivative $\partial_T \gamma^-$ (slashed contraction) given by

$$\frac{\partial \gamma^-_{\eta \sigma r}}{\partial T}(t, T) = \frac{i\Gamma_{r\sigma} e^{i\bar{\eta}\mu_r t}}{\sinh(\pi t T)} \left[ \frac{\pi t T}{\tanh(\pi t T)} - 1 \right] \tag{30}$$

$$\sim i\Gamma_{r\sigma} e^{i\bar{\eta}\mu_r t} \pi t T e^{-\pi t T} \begin{cases} \frac{1}{3} & \text{for} \quad t \ll T^{-1} \\ 2 & \text{for} \quad t \gg T^{-1} \end{cases}. \tag{31}$$

which is explicitly divergence free. As mentioned in Sec. 2, it never behaves as a power law and even vanishes identically as $T \to 0$. The second term of Eq. (29) contains the temperature derivative of the propagator $\partial_T \Pi \equiv =\!\!\!=\!\!\!\!\!\!\!\diagup\!\!\!=$. We show in App. B that this is connected to the $T$ derivative of the memory kernel via

$$\frac{\partial \Pi}{\partial T} = -i \Pi * \frac{\partial \Sigma}{\partial T} * \Pi, \tag{32}$$

where $*$ denotes time convolution. This turns Eq. (29) into a self-consistent equation for $\partial_T \Sigma$. The third term of Eq. (29) requires the temperature derivative of the effective supervertex $\partial_T G_1$. It can be obtained by differentiating (27). Here the two-point vertex $\partial_T G_{12}$ enters, which can only be expressed in an exact manner using three-point vertices $\partial_T G_{123}$ and so on. This way a hierarchy of self-consistent differential equations is obtained.

Approximations within the above general $T$-flow scheme consist in truncating this hierarchy. To do so we count the number of bare vertices present in each term, which keep track of the number of contraction functions $\gamma^-$ in which we are expanding. For example, the first and second term of Eq. (24) are counted as $\mathcal{O}(G^{+2})$ and $\mathcal{O}(G^{+4})$, respectively. In this first implementation of the method we will keep all terms in the vertex equations such that Eq. (29) consistently includes all terms of order $\mathcal{O}(G^{+6})$. This means that the $T$-flow vertex equations are

$$\partial_T G_1 = \text{[diagrams]} + \mathcal{O}(G^{+7}), \tag{33}$$

$$\partial_T G_{12} = \text{[diagram]} + \mathcal{O}(G^{+6}). \tag{34}$$

Eq. (29) together with Eqs. (33)–(34) constitute the main result of the paper. They form a closed set of self-consistent differential equations for the memory kernel $\Sigma$ and the effective supervertices $G_1$ and $G_{12}$ – describing the entire dynamics *and* transport – as function of temperature $T$.

The above derived $T$-flow is started at some high, but finite temperature $T_\infty < \infty$, where initial conditions are obtained straightforwardly using the renormalized perturbation theory: we first compute the next-to-leading order memory kernel $\Sigma(T_\infty)$ (see Eqs. (35a)–(35c) in Ref. [6]), which is then used to solve the corresponding time-nonlocal quantum master equation giving the propagator $\Pi(T_\infty)$. Inserting $\Pi(T_\infty)$ into the first two terms of Eq. (25) gives an initial value for the supervertex $G_1(T_\infty)$. Similarly the first term of Eq. (26) is used to compute $G_{12}(T_\infty)$.

Using Eq. (30) it is now easy to see that the $T$-flow reaches a fixed-point at $T = 0$ [Eq. (1)]: taking a $T$ derivative of the renormalized perturbation series (10) for $\mathcal{K}$ each summand contains exactly one $\partial_T \gamma^-_{\eta\sigma}$ factor, which vanishes for all times $t \geq 0$ as $T \to 0$ as discussed. By the decomposition (13) this implies that $\lim_{T\to 0} \partial_T \Sigma(t, T)$ vanishes.

Finally, we mention that the $t = 0$ singularity in the contraction $\gamma^-$ never contributes explicitly in Eq. (29). This is because in the first term only a non-singular slashed contraction $\partial_T \gamma^-_{\eta\sigma}$ enters. Furthermore, we show in App. C that in the second and third term the singularity is always compensated. Thus, provided the initial propagator is time-non-singular, it will remain so during the $T$-flow making the approach well behaved and suitable for numerical calculations. This is indeed the case for Anderson-like models considered here, since the initial propagator is computed using next-to-leading order renormalized perturbation theory, for which the singularity is known to be canceled out by the anticommutation of the superfermions (8) [6, 22].

## 5  Results

We now present results obtained by numerical solution of the $T$-flow equations for non-zero interaction $U$, whose implementation details are discuss in App. E. We focus on transport observables but emphasize that we have checked that every computed propagator is a completely positive map at each time $t$. This is a basic criterion for the physicality of an approximation ensuring it also properly evolves the system when it is entangled [30,38]. Furthermore, in App. F we explicitly show how the $T$-flow recovers the known exact solution at $U = 0$ [23,27,30].

### 5.1  Stationary limit

We first consider the stationary limit for the purpose of benchmarking, stressing right away that this is not the limit where a real-time formulation is supposed to be particularly advantageous. For this reason, we need to restrict our attention to bias $V \geq \Gamma$, since for smaller bias the stationary limit is reached only at relatively large times, which is of course challenging when working in the time-domain. We consider the dot at the particle-hole symmetric point $\epsilon = -U/2$ connected to two reservoirs $r \in \{L, R\}$ with temperature $T_L = T_R = T$ under a symmetric bias $\mu_{L/R} = \pm V/2$ with $\Gamma_{r\sigma} \equiv \Gamma$ independent of $r$ and $\sigma$. For sufficiently low temperatures the Kondo effect becomes important and renders both bare and renormalized perturbation theory computations unreliable.

In Fig. 1(a) we compare the obtained stationary current $I_{\text{stat}}$ as function of $V = \mu_L - \mu_R$ for $T = 0$ with results from the functional renormalization group (fRG), time-dependent density matrix renormalization group (tDMRG) and real-time quantum Monte Carlo method (QMC) reported in Refs. [8,39,40]. We find very good agreement with all four methods for $U = 2\Gamma$ and $U = 4\Gamma$. At $U = 8\Gamma$ we see that the currents predicted by our method are very close to the QMC ones, but slightly higher than the currents of fRG and tDMRG. The agreement with QMC persists for $U = 10\Gamma$ noting that for this value no fRG and tDMRG data were reported in Ref. [8].

In Fig. 1(b) we show the stationary current as function of temperature. Here each curve is efficiently obtained within a *single T*-flow renormalization group trajectory. We find that the current is monotonically increasing as $T$ is lowered. The asymptotic current for high temperatures is independent of $\epsilon$ and $U$ and given by

$$I_{\text{stat}} = \frac{\Gamma V}{4T} \quad \text{if} \quad T \gg \Gamma, \epsilon, U, V, \tag{35}$$

which can be derived from the kernels (17) and (23).

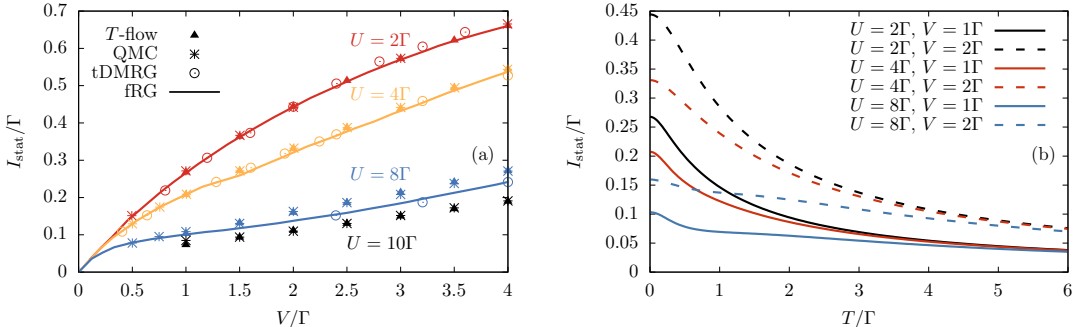

Figure 1: Stationary current at the symmetry point $\epsilon = -U/2$. (a) Comparison of the stationary current $I_{\text{stat}}$ at $T = 0$ as function of bias $V$ between the $T$-flow method, fRG, tDMRG and QMC. (b) Stationary current as function of temperature.

In Fig. 2 we show the *stationary charge fluctuations* $(\Delta n)^2_{\text{stat}} := \left\langle n^2 \right\rangle_{\text{stat}} - \left\langle n \right\rangle^2_{\text{stat}}$ . Because the stationary occupation at the symmetry point equals $\langle n \rangle_{\text{stat}} = 1$, it follows that the stationary charge fluctuations are related to the stationary occupation-correlation as

$$(\Delta n)^2_{\text{stat}} = 2\langle n_\uparrow n_\downarrow \rangle_{\text{stat}} . \tag{36}$$

The behaviour for high temperatures can be analytically calculated from Eq. (17) to be

$$(\Delta n)^2_{\text{stat}}(T) = \frac{1}{2}\left( 1 - \frac{4\epsilon + 3U}{4T} \right) \quad \text{for} \quad T \gg \Gamma, \epsilon, U, V , \tag{37}$$

which we stress also holds if the system is *not* at the particle hole symmetric point. The temperature dependence is shown in Fig. 2(a): Whereas the curves for different bias $V$ merge at high $T$ into the limiting curve (37) [inset], the fluctuations at small temperatures are suppressed as expected by Coulomb blockade. However, the fluctuations hit a global minimum at *finite T*, which is especially noticeable for small $V$, after which they increase again. For the chosen parameters this minimum occurs at $T \approx 0.4\Gamma$ for $V = \Gamma$ and then moves towards lower temperatures with increasing $V$. We attribute this enhancement of charge fluctuations at small $T$ and $V$ to the onset of the Kondo effect which in the $T$-flow method requires an account of time-correlations on increasingly larger time scales as temperature is decreased. Since a large bias suppresses the Kondo effect, the finite temperature minimum of the fluctuations should become less pronounced at larger $V$, which indeed can be seen. Fig. 2(b) shows that for the chosen parameters the fluctuations scale as

$$(\Delta n)^2_{\text{stat}}(T) = (\Delta n)^2_{\text{stat}}(T = 0)\left[ 1 - c\frac{T^2}{\Gamma^2} \right] \quad \text{if} \quad T \ll \Gamma , \tag{38}$$

where the constant $c$ depends on $U$ and $V$. This $T^2$ scaling is ubiquitous for the Kondo effect in the low temperature Anderson model and appears for example also in the conductance as function of $V$ or $T$ (for $V, T \lesssim T_K$) [17, 41].

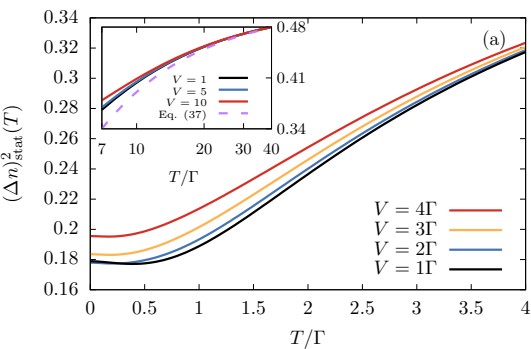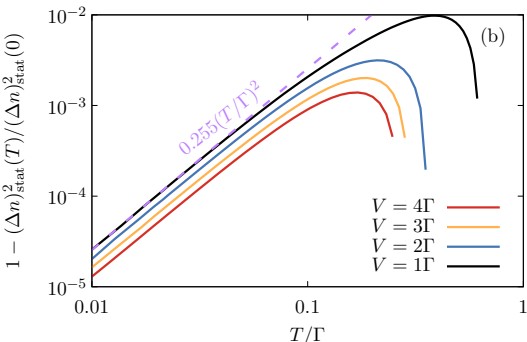

Figure 2: Stationary charge fluctuations $(\Delta n)^2_{\text{stat}}$ as function of temperature for $U = 8\Gamma$, $\epsilon = -U/2$ and several bias voltages. (a) Low and intermediate temperature regime. Inset: High temperature regime. (b) Scaling of the fluctuation for $T \ll \Gamma$.

## 5.2 Transient effects

We now turn to the transient dynamics. Here we find quite generally that the short time behaviour of the propagator is independent of temperature. This contribution stems not just from the leading-order infinite-temperature limit, but additionally from the initial temperature-independent part the memory kernel: For small times $\delta t$ we have

$$\Pi(\delta t) = \Pi_\infty(\delta t) - \frac{i}{2}\Sigma(t=0)\delta t^2, \tag{39}$$

where for conciseness we have not expanded the first term, with the temperature-independent zero-time kernel

$$-i\Sigma(t=0) = \frac{2}{\pi}\Gamma G_1^+ L_\infty G_{\bar{1}}^+ + \frac{2}{\pi}\Gamma \sum_{r\sigma}\mu_r G_{+\sigma}^+ G_{-\sigma}^+. \tag{40}$$

Here the second term does not contribute for symmetric bias $\mu_L = -\mu_R$ considered here. This $T$-independence means that there is no $T$-flow of the propagator at short times $\delta t \ll \Gamma^{-1}$. As a consequence all local observables are initially insensitive to temperature as, for example, the occupations

$$\langle n_\sigma\rangle(\delta t) = \frac{1}{2} - e^{-2\Gamma\delta t}\left(\frac{1}{2} - \langle n_\sigma\rangle_{\rho_0}\right) + \left[U\left(\frac{1}{2} - \langle n_{\bar{\sigma}}\rangle_{\rho_0}\right) - \frac{U+2\epsilon}{2}\right]\frac{\Gamma}{\pi}\delta t^2, \tag{41}$$

where again we do not expand the exponential for conciseness. Here the first two terms describe decay to half filling coming from the infinite temperature propagator in (39). The third and fourth terms add quadratic corrections depending on the initial deviation of $n_{\bar{\sigma}}$ from half filling and the level deviation from the symmetry point. This is shown in Fig. 3, where the transient occupation $\langle n\rangle(t) = \langle n_\uparrow\rangle(t) + \langle n_\downarrow\rangle(t)$ is plotted for several temperatures. Because the initial evolution is $T$-independent as explained above and the stationary occupation is fixed by the particle-hole symmetry, temperature can only affect the intermediate occupations. A noticeable detail of this crossover from weak to strong coupling is that for this moderate value of the interaction the occupation slightly overshoots its stationary value $\langle n\rangle_{\text{stat}} = 1$ for the lowest temperatures, but this effect is lost already for $U = 8\Gamma$ (not shown).

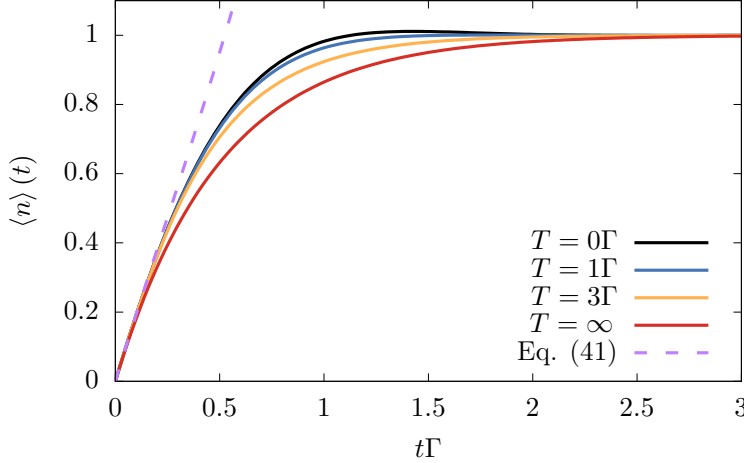

Figure 3: Transient occupation $\langle n\rangle(t)$ for $U = 4\Gamma$, $\epsilon = -U/2$, bias $V = \Gamma$ and several temperatures. Initially the dot is empty, $\rho_0 = |0\rangle\langle 0|$. We note that the $\mathcal{O}(\delta t^2)$ contributions of Eq. (41) are negligible here, but can play a role, see Fig. 5.

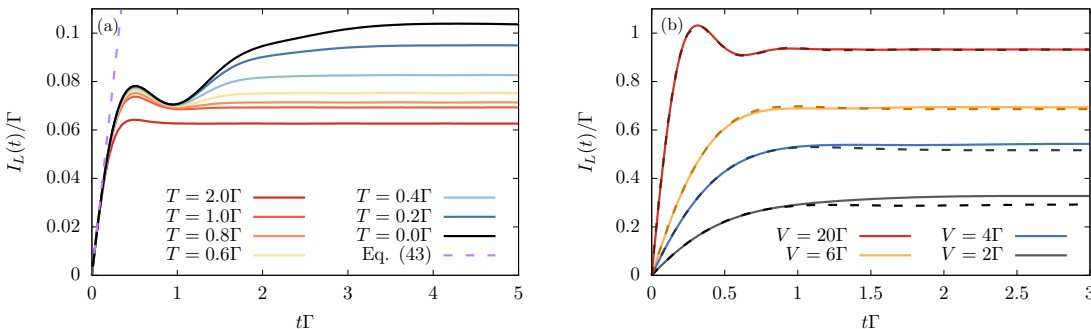

Figure 4: (a) Transient current $I_L(t)$ for $U = 8\Gamma$, $\epsilon = -U/2$, $V = \Gamma$ and several temperatures. Initially the dot is singly occupied, $\rho_0 = |\uparrow\rangle\langle\uparrow|$. (b) Transient current $I_L(t)$ for $U = 4\Gamma$, $\epsilon = -U/2$, $T = 0.1\Gamma$ and several bias voltages. Solid lines: $T$-flow. Dashed lines: 2PI approach from Ref. [42].

Interestingly, for non-local observables the short-time behaviour may also be temperature independent. An example is the particle current plotted in Fig. 4. Similar to the decomposition (39) there are contributions from both the infinite temperature current and the $T$-independent zero-time current-kernel, e.g., for the left reservoir:

$$I_L(\delta t) = I_{L,\infty}(\delta t) - i\operatorname{Tr}\Sigma_{I_L}(t=0)\rho_0\delta t \tag{42}$$

$$= (V - U - 2\epsilon)\frac{\Gamma\delta t}{\pi} + (1 - \langle n\rangle_{\rho_0})\Gamma\left[1 + (U - 2\pi\Gamma)\frac{\delta t}{\pi}\right]. \tag{43}$$

Again we stress that this result also holds if the system is *not* at the particle hole symmetry point. In Fig. 4(a) $I_L(t)$ is shown for large interaction $U = 8\Gamma$ at bias $V = \Gamma$ as the temperature is lowered. As expected the initial onset of the current follows Eq. (43). Whereas for $T \gtrsim \Gamma$ the current monotonically converges to its stationary value, at lower temperatures the current after an initial increase first decreases until $t\Gamma \approx 1$ and then turns up again. For $T \lesssim 0.2\Gamma$ the stationary current is significantly higher than the local maxima at $t\Gamma \approx 1/2$. The local minimum at $t\Gamma \approx 1$ becomes less pronounced if $U$ is decreased and eventually vanishes (not shown).

In Fig. 4(b) we compare the transient currents obtained by the $T$-flow with those obtained in Ref. [42] using a two-particle-irreducible effective action (2PI) approach at low temperature $T = 0.1\Gamma$ and intermediate interaction $U = 4\Gamma$. We find overall good agreement. In particular, both predict that the current slightly overshoots its stationary value at large bias. At small bias the stationary current of the 2PI approach is slightly smaller compared to our $T$-flow result, which in the stationary benchmarks in Fig. 1(a) compared favourably with other methods.

Finally, as an application we consider the transient effects of the interaction in the empty-orbital regime $\epsilon \gg \{\Gamma, V\}$, which is characterized by $\langle n\rangle_{\text{stat},T=0} \approx 0$. Perhaps surprisingly, preparing the dot in a state with a higher occupation than its stationary occupation need not lead to a simple decay of the occupation. Instead, it is possible that the dot initially fills up *more* as predicted in Ref. [30] on quite general grounds. How this can happen can be understood specifically from Eq. (41), which shows that initially the occupation grows towards half filling – away from the stationary value – as dictated by $\Pi_\infty$. Indeed, in Fig. 5 the occupation initially increases until $t\Gamma \approx 0.3$, after which the naively expected monotonic decay starts. The occupations then reenter their initial value precisely at the reentrant time $t_r = \Gamma^{-1}$. More strongly, for the chosen initial state this reentrance occurs for *any local observable* of the dot, as for example the correlation in Fig. 5, implying that the entire reduced density operator returns to its initial value.

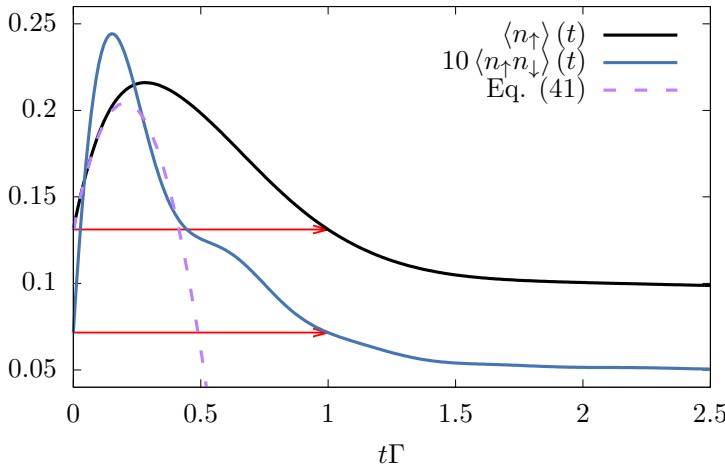

Figure 5: Reentrant effect for dot occupation and correlation for parameters $U = 8\Gamma$, $\epsilon = 2.75\Gamma$, $V = \Gamma$ at $T = 0$. The reentrant effect for time $t_r = \Gamma^{-1}$ is realized by the initial state $\rho_0$ with $\langle n_\uparrow \rangle_{\rho_0} = \langle n_\downarrow \rangle_{\rho_0} \approx 0.131$ and $\langle n_\uparrow n_\downarrow \rangle_{\rho_0} \approx 0.007$.

This at first puzzling reentrant behaviour was already explained in Ref. [30] in general terms showing that it is generically enforced in non-semigroup dynamics by the fundamental property of trace-preservation of the dynamics. In short, for every time $t_r$ the propagator has a fixed point, i.e., some operator denoted $\rho_1(t_r)$ such that $\Pi(t_r)\rho_1(t_r) = \rho_1(t_r)$, which is moreover a physical state, $\rho_1(t_r) \geq 0$ and $\mathrm{Tr}\,\rho_1(t_r) = 1$. Thus, all local observables must return to their initial values at time $t_r$ if the initial state $\rho_0 = \rho_1(t_r)$ is prepared. (Note that this argument does not imply reentrant behaviour of non-local observables such as currents measured outside the system.) Whereas in Ref. [30] this general effect was predicted, it was illustrated only for the occupation of a non-interacting spinless quantum dot coupled to a single reservoir, a solvable model. Here we have illustrated that it occurs for more than one observable and shown that it remains clearly visible in the strongly-interacting, low-temperature case under finite bias transport conditions. We highlight that the rationale behind the $T$-flow method ties in directly with the competition between finite-$T$ and infinite-$T$ dynamics that drives this physical effect.

# 6 Summary and outlook

In this paper we presented the $T$-flow renormalization group method, which uses the physical environment temperature to achieve a flow of the density-operator dynamics to its nontrivial low-temperature limit. Starting from the simple high-temperature limit, the temperature is lowered in many small steps using the self-consistent RG equations (29)–(34). In this way we collect useful information about the physics at *all* traversed finite temperatures, which sets our scheme apart from RG methods using unphysical flow parameters.

We implemented the RG equations directly in time space at the example of an interacting Anderson dot including vertex corrections. For voltages on the order of the coupling or larger, stationarity is reached quickly, allowing us to benchmark our transient method in the stationary regime. We demonstrated quantitative agreement in the current-voltage characteristics, noting in particular that the agreement with accurate quantum Monte Carlo simulations extends up to $U = 10\Gamma$. Comparing transient currents with the 2PI Green's function results we found good agreement as well. Interestingly, we could show analytically that the short

time dynamics of both local and nonlocal observables are temperature independent in the wide-band limit considered here, with important contributions from the short-time memory kernels. The observed collapse of the data onto a $T$-independent limiting curve may be of interest for experimental studies of transient transport.

As an application we investigated the reentrant effect found in Ref. [30] for a solvable non-interacting model: the charge prepared on a quantum dot which is destined to decay can initially show an unexpected pronounced further accumulation. We showed that this effect is due to a generic interplay between short-time infinite-$T$ dynamics and long-time finite-$T$ corrections and remains clearly visible even for strong interactions and finite-bias non-equilibrium. Moreover, we illustrated that this effect does not only occur for the occupation but also for the *correlation* of two electrons in agreement with general arguments about non-semigroup dynamics put forward in Ref. [30].

For the formulation and application of the method we focused on the case of equal reservoir temperatures and studied the transient approach to non-equilibrium stationary states and transport quantities. However, we also provided the general formulation for distinct temperatures of the reservoirs whose application to thermoelectric transport with strong coupling and interaction is interesting. The additional heat currents of interest in this situation [43, 44] can be computed by straightforward extension of the presented technique for the charge current. Moreover, the presented method can be extended in several further directions: (i) Since the $T$-flow allows to avoid the frequency domain it is straightforward to include non-periodic driving of bias and gate voltages [$V(t)$ and $\epsilon(t)$] or tunnel barriers [$\Gamma(t)$]. This comes at the numerical price of dealing with double (triple) time-dependence of the contraction functions $\gamma_1^p$ and the memory kernel $\mathcal{K}$ (the supervertex $G_1$), but should present no principal problem. (ii) However, it is equally well possible to formulate the $T$-flow entirely in the Laplace-frequency domain by changing the translation rules of the diagrams as in Ref. [3, 22]. This may provide more direct access to stationary quantities than by converging transient calculations well into the stationary regime, in particular for regimes where $T, V \ll \Gamma$. How this compares with the $E$-flow scheme is an interesting open question. (iii) The $T$-flow can also address systems with bosonic environments provided a renormalized perturbation theory around the infinite temperature and wideband limit is set up analogous to fermionic environments [20, 45], which seems possible.

We emphasized throughout that the presented $T$-flow scheme is naturally suggested by physical considerations underlying the renormalized perturbation expansion. Nevertheless, further considerations of its physical underpinnings would be of interest. Indeed, early time-domain formulations of the density-operator RTRG [45, 46] were motivated by considerations similar to those given in Sec. 2, but encountered technical issues. These were initially resolved by reformulations which abandoned the time-domain [47, 48] and started from a renormalized perturbation theory which was later identified as describing the $T = \infty$ limit [22, 23] as used here. This renormalization by the $T = \infty$ reference solution plays a crucial role in the construction of a well-defined RG flow for the open-system dynamics, in particular, for the special treatment required for the stationary state. This is problematic when starting from a standard bare perturbation expansion ("zero eigenvalue problem", see the discussion preceding Eq. (199) in Ref. [3]).

In the present paper we instead returned to the time-domain by exploiting the insights gained in the above cited works. This has the advantage that the similarities and differences with Wilson's RG [32] become apparent, in particular the focus on the long-range correlations which become explicit in the renormalized perturbation theory about $T = \infty$. The ordinary perturbation expansion does not reveal the relevant correlations since the $T = \infty$ and $T < \infty$ contributions are completely mixed up [22, 23]. This constitutes a key difference of the $T$-flow to other RG approaches. Its apparently successful application here warrants

further, more detailed consideration of these physical underpinnings, in particular its close connection to the *generation of memory effects*. Importantly, here memory is characterized as retardation [5], which is related to but not the same as non-divisibility of dynamical maps [31] ("non-Markovianity").

The return to the time-domain achieved by our work may also enable general physical arguments to be applied more easily to technical considerations, especially since our RG flow parameter can in principle be changed physically in the lab. A $T$-flow step establishes a mapping between two *entire physical evolutions* at adjacent temperatures. Operationally this corresponds to an intervention on the initially decoupled reservoir state $\rho_R(T) \to \rho_R(T - \delta T)$ [Eq. (5) ff.] before the interaction with the system is started and before the environment is discarded (integrated out). It has been stressed [31] that interventions on the environment represent yet another way of characterizing memory effects beyond the framework of retardation and non-divisibility of the reduced evolution mentioned above. Operationally defined mappings of entire evolutions have been studied in detail in quantum information using the supermap [49] and process-tensor formalisms [50] and these may find new applications here. To apply such considerations here it is a crucial advantage that the $T$-flow allows one to stay in the time-domain, since transformation to frequency domain tends to complicate rather than simplify operational properties of time-evolutions [38].

# Acknowledgements

We thank V. Bruch and M. Pletyukhov for valuable discussions and H. Schoeller for inspiring lectures on RG methods. We thank T. Gasenzer for useful correspondence regarding Ref. [42].

**Funding information** K.N. acknowledges support by the Deutsche Forschungsgemeinschaft (RTG 1995).

# A  Definition of effective supervertices

Here we give the precise definition of the supervertices and show how the vertex diagrams can be translated into explicit equations. First, it is important to keep in mind that in contrast to the bare superfermion $G_1^+$ [Eq. (8)] the effective supervertex $G_1$ also has a dependence on time. Specifically we need to distinguish the time arguments of the latest, the uncontracted and the earliest vertex within $G_1$, which we label as $t$, $\tau$ and $s$ respectively. Thus $t \geq \tau \geq s$ and we denote $G_1 = G_1(t, \tau, s)$. For time translation invariant systems this simplifies to $G_1(t-\tau, \tau-s)$. Every vertex *except the uncontracted one* is associated with a prefactor of $-i$, and every cut contraction line with a fermion minus sign. Importantly the contraction function associated with the uncontracted vertex is not part of $G_1$ itself. Thus, the first two diagrams in the definition of $G_1$ [Eq. (25)] are translated as

$$\text{(diagram)} = G_1^+ \bar{\delta}(t-\tau)\bar{\delta}(\tau-s), \tag{44}$$

$$\text{(diagram)}_{t\ \tau\ s} = \gamma_2^-(t-s)G_2^+\Pi(t,\tau)G_1^+\Pi(\tau,s)G_2^+, \tag{45}$$

where we indicated the time arguments in the second diagram. Higher order terms also contain internal vertices with time arguments labelled from left to right as $t_1 > \cdots > t_n$ over

which one has to integrate in a time-ordered way. For example, the third term of (25) reads

$$
\begin{aligned}
\text{[diagram]} = \int_{\tau}^{t} dt_1 \int_{\tau}^{t_1} dt_2 \gamma_2^-(t-t_2)\gamma_3^-(t_1-s) \\
\times G_2^+\Pi(t,t_1)G_3^+\Pi(t_1,t_2)G_{\bar{2}}^+\Pi(t_2,\tau)G_1^+\Pi(\tau,s)G_{\bar{3}}^+.
\end{aligned} \tag{46}
$$

The 2-point vertex $G_{12} = G_{12}(t,\tau_1,\tau_2,s)$ is defined such that the uncontracted vertex with index 1 (at time $\tau_1$) is always to the left of the uncontracted vertex with index 2 (at time $\tau_2$), i.e., $\tau_1 > \tau_2$. Thus

$$
\text{[diagram]} = -\gamma_3^-(t-s)G_3\Pi(t,\tau_1)G_1\Pi(\tau_1,\tau_2)G_2\Pi(\tau_2,s)G_{\bar{3}}. \tag{47}
$$

## B    Temperature dependence of the propagator $\Pi$

The propagator $\Pi$ can be computed from the renormalized kernel $\Sigma$ via the Dyson equation

$$
\Pi(t) = \Pi_\infty(t) - i[\Pi_\infty * \Sigma * \Pi](t), \tag{48}
$$

where $*$ denotes time convolution. Suppressing time arguments and taking a derivative with respect to the temperature $T_r$ of the $r$-th reservoir it follows that

$$
\frac{\partial \Pi}{\partial T_r} = \Pi_\infty * \frac{\partial [-i\Sigma]}{\partial T_r} * \Pi + \Pi_\infty * [-i\Sigma] * \frac{\partial \Pi}{\partial T_r} \tag{49}
$$

$$
= -i\Big(\Pi_\infty + \Pi_\infty * [-i\Sigma] * \Pi_\infty + \cdots\Big) * \frac{\partial \Sigma}{\partial T_r} * \Pi \tag{50}
$$

$$
= -i\Pi * \frac{\partial \Sigma}{\partial T_r} * \Pi. \tag{51}
$$

To obtain Eq. (50) one iterates the self-consistent equation Eq. (49) for $\partial_T \Pi$ treating $\Pi$, $\Pi_\infty$ and $\Sigma$ as given. Eq. (51) follows by recognizing the term in parenthesis as the solution of the self-consistent equation (48) for $\Pi$. Setting $T$ equal for all reservoirs we obtain Eq. (32) of the main text. Inserting the leading order term of $\partial_T \Sigma$ it is straightforward to show that the leading short-time dependence of $\partial_T \Pi$ is *quartic* for small times $\delta t$:

$$
\frac{\partial \Pi}{\partial T_r}(\delta t) = -\frac{\pi}{36} T_r \Gamma_{r\sigma_1}\Big(G_1^+ L_\infty G_{\bar{1}}^+ + \eta_1 \mu_r G_1^+ G_{\bar{1}}^+\Big)\delta t^4. \tag{52}
$$

This explains the $T$-independent short-time behaviour discussed in the main text [Eq. (41)].

## C    Finiteness of $T$-flow equations

Here we establish that the $T$-flow equations are free of time-divergences. The diagrams in Eq. (29) are explicitly given by

$$
\text{[diagram]}(t,s) = -\int_{s}^{t} dt_1 \int_{s}^{t_1} dt_2 \frac{\partial \gamma_1}{\partial T}(t-t_2)G_1^+\Pi(t,t_1)G_{\bar{1}}(t_1,t_2,s), \tag{53}
$$

$$
\text{[diagram]}(t,s) = -\int_{s}^{t} dt_1 \int_{s}^{t_1} dt_2 \gamma_1(t-t_2)G_1^+\frac{\partial \Pi}{\partial T}(t,t_1)G_{\bar{1}}(t_1,t_2,s), \tag{54}
$$

$$
\text{[diagram]}(t,s) = -\int_{s}^{t} dt_1 \int_{s}^{t_1} dt_2 \gamma_1(t-t_2)G_1^+\Pi(t,t_1)\frac{\partial G_{\bar{1}}}{\partial T}(t_1,t_2,s). \tag{55}
$$

Whereas (53) does not contain any singularities, this is not immediately obvious for Eqs. (54) and (55). In Eq. (54) we use that $\partial_T \Pi(t) = \mathcal{O}(t^4)$, see Eq. (52). This small-time behaviour regularizes the $1/t$ divergence of the contraction function. The finiteness of (55) can be seen by switching the order of integrations:

$$\int_s^t dt_1 \int_s^{t_1} dt_2 \gamma_1(t - t_2) = \int_s^t dt_2 \int_{t_2}^t dt_1 \gamma_1(t - t_2). \tag{56}$$

Now the inner $t_1$ integral vanishes as $\mathcal{O}(t - t_2)$ making the term finite. Using very similar arguments one establishes that all diagrams in (33) and (34) are well behaved, making the $T$-flow equations explicitly time-singularity free.

## D  Different reservoir temperatures

Here we show how the $T$-flow as presented in the main text can be generalized to the case where each reservoir has a different temperature $T_r$. This can be applied in various ways. For this discussion it is useful to collect all temperatures into a single vector $\vec{T} = (T_1, T_2, \ldots, T_n)$. The $T$-flow is then started at high, but finite temperatures $\vec{T}_\infty = (T_{\infty,1}, T_{\infty,2}, \ldots, T_{\infty,n})$, allowing us to compute an accurate initial condition using the renormalized perturbation theory. We next chose a path $\vec{T}(\alpha) = (T_1(\alpha), T_2(\alpha), \ldots, T_n(\alpha))$ in this $n$ dimensional temperature space parametrised by $\alpha : 0 \to 1$, passing through the temperature-biased configurations of interest. If we denote by $\vec{T}_0$ the final target configuration of the reservoir temperatures (in the main text $\vec{T}_0 = \vec{0}$) then $\vec{T}(\alpha = 0) = \vec{T}_\infty$ and $\vec{T}(\alpha = 1) = \vec{T}_0$. The case from the main text, where all reservoirs are cooled at the same rate, corresponds to $\vec{T}(\alpha) = \vec{T}_\infty + \alpha(\vec{T}_0 - \vec{T}_\infty)$. Alternatively we could keep $T_2, \ldots, T_n$ fixed while cooling $T_1$, and afterwards cool $T_2$ etc..

To generalize the $T$-flow equations we replace all derivatives

$$\partial_T \to \partial_\alpha = \partial \vec{T} / \partial \alpha \cdot \nabla_{\vec{T}}, \tag{57}$$

in Eqs. (29)–(34). For example, slashed contractions now denote

$$\frac{\partial \gamma_{\eta\sigma}^-}{\partial \alpha} = \sum_r \frac{\partial T_r}{\partial \alpha} \frac{i\Gamma_{r\sigma} e^{i\bar{\eta}\mu_r t}}{\sinh(\pi t T_r)} \left[ \frac{\pi t T_r}{\tanh(\pi t T_r)} - 1 \right]. \tag{58}$$

Note that the slashed propagator is now given by the key relation

$$\partial_\alpha \Pi = -i\Pi * \partial_\alpha \Sigma * \Pi, \tag{59}$$

[cf. (32)]. With these conventions the same diagrammatic rules apply, which makes the implementation of the $T$-flow for distinct temperatures straightforward. Finally, we point out that closed temperature loops have no thermodynamic meaning here, because we are not lowering temperature in time (each RG step computes an entire evolution).

## E  Numerical solution of the $T$-flow equations

The truncated $T$-flow equations from the main text form a closed set of implicit (self-consistent) differential equations for $\Sigma$, $G_1$ and $G_{12}$ and we here comment on their numerical discretiza-

tion. Suppressing time arguments these equations are of the form

$$\frac{\partial \Sigma}{\partial T} = \mathcal{F}_0\left[\Sigma, \frac{\partial \Sigma}{\partial T}, G_1, \frac{\partial G_1}{\partial T}\right], \tag{60}$$

$$\frac{\partial G_1}{\partial T} = \mathcal{F}_1\left[\Sigma, \frac{\partial \Sigma}{\partial T}, G_1, \frac{\partial G_1}{\partial T}, G_{12}, \frac{\partial G_{12}}{\partial T}\right], \tag{61}$$

$$\frac{\partial G_{12}}{\partial T} = \mathcal{F}_2\left[\Sigma\right], \tag{62}$$

where the functionals $\mathcal{F}_i$ are given by the right-hand sides of Eqs. (29), (33) and (34). We do not explicitly indicate the dependence on the propagator $\Pi$ and its temperature derivative $\partial_T \Pi$, since these can be computed using $\Sigma$ and $\partial_T \Sigma$ as the solutions of Eqs. (6) and (32), respectively. Defining the vector $\Phi := (\Sigma, G_1, G_{12})$ the $T$-flow equations thus have the form

$$\frac{\partial \Phi}{\partial T} = \mathcal{F}\left[\Phi, \frac{\partial \Phi}{\partial T}\right]. \tag{63}$$

To simplify the discussion we assume an equidistant temperature grid $T_n := n\delta T$ for some small stepsize $\delta T > 0$, noting that in practice the stepsize should be varied based on local error estimates to reduce numerical effort and improve accuracy. Our goal is to compute $\Phi_n := \Phi(T_n)$ and $\partial_T \Phi_n := \partial_T \Phi(T_n)$ on this grid assuming $\Phi_{n+1}, \Phi_{n+2}, \dots$ and $\partial_T \Phi_{n+1}, \partial_T \Phi_{n+2}, \dots$ are already available. To do so we first approximate $\partial_T \Phi_n \approx (-3\Phi_n + 4\Phi_{n+1} - \Phi_{n+2})/(2\delta T)$ leading to

$$\Phi_n = \frac{4}{3}\Phi_{n+1} - \frac{1}{3}\Phi_{n+2} - \frac{2}{3}\mathcal{F}\left[\Phi_n, \frac{-3\Phi_n + 4\Phi_{n+1} - \Phi_{n+2}}{2\delta T}\right]\delta T + \mathcal{O}(\delta T^3). \tag{64}$$

This is the well-known second order backwards differentiation formula (BDF2) [51]. To evaluate Eq. (64) further we approximate on the right hand side $\Phi_n = \Phi_n^{(P)} + \mathcal{O}(T^3)$, where $\Phi_n^{(P)}$ denotes the Adams-Bashforth predictor [51]

$$\Phi_n^{(P)} := \Phi_{n+1} - \left(\frac{3}{2}\frac{\partial \Phi}{\partial T}\bigg|_{n+1} - \frac{1}{2}\frac{\partial \Phi}{\partial T}\bigg|_{n+2}\right)\delta T. \tag{65}$$

Thus we obtain both $\Phi_n$ and $\partial_T \Phi_n = \mathcal{F}\left[\Phi_n^{(P)}, (-3\Phi_n^{(P)} + 4\Phi_{n+1} - \Phi_{n+2})/(2\delta T)\right]$ as wanted.

# F   Exact solution at $U = 0$

Here we show how the $T$-flow equations (29) and (32)–(34) recover the exact solution for the non-interacting spin-degenerate Anderson dot. The main simplification in this case is that all terms with more than four creation superfermions vanish for algebraic reasons as discussed in Ref. [6, 23, 30]. Therefore the infinite $T$-flow hierarchy terminates at finite order and is completely contained in the contributions of the main text. In fact they simplify to

$$-i\frac{\partial \Sigma}{\partial T} = \quad + \quad + \quad , \tag{66}$$

$$\quad = \quad . \tag{67}$$

In the second term of Eq. (66) we can use a bare vertex (instead of an effective one) and in the third term a bare propagator $\Pi_\infty$ (instead of a full one) because the corrections to this are of order $\mathcal{O}(G^{+6})$ and vanish algebraically. For the same reason Eq. (67) only contains bare vertices and propagators. With the initial condition $G_1(T = \infty, t - \tau, \tau - s) = G_1^+ \bar{\delta}(t - \tau)\bar{\delta}(\tau - s)$

for the supervertex we can immediately integrate (67), since only the contraction depends on temperature:

$$\text{(diagram)} = \text{(diagram)} + \text{(diagram)}. \tag{68}$$

Note that the first term is equal to the initial condition and contains two $\bar{\delta}$ functions of time not indicated diagrammatically. Insertion of $G_1$ [Eq. (68)] and $\partial_T G_1$ [Eq. (67)] into the first and last term of (66) respectively, gives

$$-i\frac{\partial \Sigma}{\partial T} = \text{(diagram)} + \text{(diagram)} + \text{(diagram)} + \text{(diagram)} \tag{69}$$
$$= \text{(diagram)} + \text{(diagram)} + \text{(diagram)} + \text{(diagram)} + \text{(diagram)},$$

where in the third term of the first line we again replaced a full propagator by a bare one. In the last line we inserted the expansion of the full propagator and its temperature derivative [Eq. (32) with (66)] using that all orders greater than $\mathcal{O}(G^{+4})$ vanish algebraically. With the initial condition $\Sigma(T = \infty) = 0$ we can integrate the last equation and recover the exact memory kernel for the $U = 0$ Anderson dot

$$-i\Sigma = \text{(diagram)} + \text{(diagram)} + \text{(diagram)}. \tag{70}$$

This is the result computed in Ref. [23] [Eq. (123), Sec. 4 and App. F loc. cit.], where the full solution is analysed in detail, see also Ref. [30].

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
