# Peer review of "Renormalization group for open quantum systems using environment temperature as flow parameter"

_SciPost Physics, doi:SciPost Phys. 12, 121 (2022)_

## Round 1 · Referee Report · Anonymous (Referee 1) · 2021-12-24

Strengths

1 - Interesting generalization of fRG ideas, allowing one to study transient phenomena
2 - Clear presentation of the method and thorough analysis of the model
3 - Synergy of analytics and numerics

Weaknesses

1 - A connection of the proposed scheme to other RG approaches could be described in a more extensive way
2 - Consideration of the paper appears too formal; a discussion in more physical (qualitative) terms should be extended
3 - The method is initially formulated for the general case of distinct temperatures of reservoirs; however, all the results are given for the case of equal temperatures

Report

This is a nice attempt to formulate a renormalization-group approach to interacting systems by explicitly using the physical temperature as a flowing scale. The authors applied the developed machinery to the Anderson model of an interacting quantum dot and demonstrated a quantitative agreement between their results and those obtained by other numerical methods.

Let me comment on the points that could be improved in the manuscript.

  1. A connection of the proposed scheme to other RG approaches could be described in a more extensive way. Indeed, usually temperature serves as an infrared cutoff for the running energy scale, e.g., in a poor-man scaling approach. It would be interesting to see this connection on a formal level, comparing the "conventional" RG approach with temperature serving only as a cutoff with the framework developed here. For this type of comparison, it would be nice to see a comparison of the approaches at an analytical level; for example, a derivation of the Luttinger-liquid renormalization of the finite-temperature conductance through a barrier (either in the limit of a weak barrier, or in the limit of weak interaction) in a quantum wire would be very instructive.

  2. Consideration of the paper appears too formal; a discussion in more physical (qualitative) terms should be extended. It is well known that, in interacting systems, temperature separates the energy domains where the physics is dominated by real processes (with energy transfers smaller than T) or by virtual processes (energy transfers larger than T). The latter usually yield renormalization of the quantities entering the effective kinetic equation. In the present framework, it is not clear whether only the latter processes are accounted for, or the real inelastic processes are also effectively captured by the developed method. Further, it is not quite clear whether the procedure depends on the initial density matrix of the system and how equilibration processes that lead to thermalization of the system state are described in this approach. I strongly suggest the authors discussing such points in a qualitative manner and simple physical terms.

3 . The method is initially formulated for the general case of distinct temperatures of reservoirs; however, all the results are given for the case of equal temperatures. It is extremely interesting to see how the method is applied to a situation with different temperatures of the reservoirs, which would correspond to a non-equilibrium steady state of the system. In particular, whether the notion of "non-equilibrium dephasing" (see, e.g., papers by Gutman et al. on non-equilibrium bosonization) would naturally emerge in this setting. But even without performing the numerical analysis of the flow equations in this intriguing case, it would be nice to present the set of such equations explicitly.

Requested changes

1 - Extend the discussion of a relation between the proposed approach and other RG approaches with temperature serving as an infrared cutoff; present an analytical solution of the derived flow equations in some tractable well-known model.

2 - Add a discussion of the approach in more qualitative terms (see report).

3 - Present the flow equations for the general case of non-equal temperatures of the reservoirs.

  • validity: high
  • significance: high
  • originality: high
  • clarity: high
  • formatting: excellent
  • grammar: excellent

Author:  Konstantin Nestmann  on 2022-03-17  [id 2292]

(in reply to Report 1 on 2021-12-24)
Category:
answer to question
pointer to related literature

We thank the referee for the interesting questions and suggestions in the positive report. We address them below in the order of the requested changes and append a remark at the end. We have supplied a color-coded PDF in which all changes have been marked including various small additional improvements and clarifications. In further extending our numerical results we noted that Fig. 2b contained curves which were insufficiently converged. This has been corrected without affecting any part of the text and conclusions.

$1$. Extend the discussion of a relation between the proposed approach and other RG approaches with temperature serving as an infrared cutoff; present an analytical solution of the derived flow equations in some tractable well-known model.

A connection of the proposed scheme to other RG approaches could be described in a more extensive way.

For this type of comparison, it would be nice to see a comparison of the approaches at an analytical level;

Indeed, usually temperature serves as an infrared cutoff for the running energy scale, e.g., in a poor-man scaling approach.It would be interesting to see this connection on a formal level, comparing the "conventional" RG approach with temperature serving only as a cutoff with the framework developed here.

Revision To accomodate the referee's suggestion we have inserted the new Appendix F "Exact solution for $U=0$" in the revised manuscript. There we explain how the exact solution of the Anderson model for $U=0$ (spin-degenerate resonant level) is obtained within the presented $T$-flow approach.

However, there is no obvious way of comparing with methods using a different flow parameter (other than comparing final results as we already do in the benchmark tests in Section 5 of the original manuscript). Indeed, the referee correctly points out the key complication in finding such a simple relation between our $T$-flow and "conventional" RG methods: temperature is a flow parameter and cannot serve as -or be compared to- a cut-off scale. As we argue further in point 2 below one may wonder whether attempting to find such a relation is necessary for physical understanding, since the $T$-flow method already has a quite clear motivation in close analogy to Wilson's arguments as described in Section 2 of the manuscript.

Revision We have added a paragraph at the end of Section 2 discussing the above understandable concerns of the referee. Furthermore, in the Summary and Outlook Section 6 we have extended the discussion of the most closely related RG methods which preceded the $T$ flow.

$2$. Add a discussion of the approach in more qualitative terms (see report).

We agree that further considerations of the physical underpinnings of the $T$-flow would be of interest as we stated in the Outlook Section 6. The referee inquires about various particular points:

(A) Elastic/inelastic contributions

It is well known that, in interacting systems, temperature separates the energy domains where the physics is dominated by real processes (with energy transfers smaller than T) or by virtual processes (energy transfers larger than T). [...] In the present framework, it is not clear whether only the latter [virtual elastic] processes are accounted for, or the real inelastic processes are also effectively captured by the developed method

Consideration of the paper appears too formal; a discussion in more physical (qualitative) terms should be extended.

I strongly suggest the authors discussing such points in a qualitative manner and simple physical terms.

Before attempting to reformulate the issue addressed by the referee and answer the question we need to explain why the particular "simple physical terms" that the referee suggests are not applicable here:

(a) In the ordinary bare perturbation expansion in the tunnel coupling, i.e., around the limit where system and reservoir are decoupled, the "energy transfers" mentioned by the referee can be discussed (but already involve subtleties) in the context of quantum dot transport. One standardly distinguishes for example (see for a review R. Gaudenzi et al, J. Chem. Phys. 146, 092330 (2017)):

  • Order $\Gamma$ contributions ("single-electron tunneling") which are significant only when the energy transfered to the dot matches the energy coming from the reservoirs as in Fermi's Golden Rule. (Already this is a subtle matter since it is well established that this does not hold for additional "Lamb-shift" terms which are not captured by the Golden Rule but arise in the same order $\Gamma$ (!) and can significantly contribute [theory: König, Martinek, Phys. Rev. Lett. 90, 166602 (2003), experiment: Hauptmann et. al. Nature Physics 4, 373 (2008)].

  • Order $\Gamma^2$ processes ("cotunneling") are denoted "(in)elastic" when energy transfer between reservoir and dot is smaller (larger) than temperature. (Order $\Gamma^2$ Lamb-shift contributions further complicate matters, [Misiorny, et al, Nature Physics 9, 801, (2013)] since the dot + reservoir energy are no longer conserved due to the significant coupling. The total energy including the coupling is conserved.)

We are a bit confused that the convention mentioned by the referee is opposite the second standard convention. (We suspect this may relate to the difference between the picture of a 1 dim. wire and 0 dim. system coupled to reservoirs, see our comment at the end.)

(b) Crucially, in the present paper we do not start from this bare perturbation theory in $\Gamma$. This means that the above standard distinctions can no longer be made:

  • We start from the exact solution for $T=\infty$ accounting for all tunneling contributions in this limit: the dot and the reservoir are coupled with finite energy $\Gamma$ and the reservoir is traced out (not: decoupled!). This means that contrary to case (a) the energy of the dot is not conserved by the reference evolution and is meaningless (the system is open). The time-evolution is governed by the renormalized Liouvillian (15) which is not a commutator of an energy operator. (One can thus not even say that "all energy transitions are smaller than the infinite thermal energy" since there is no dot energy associated with the time evolution.)

  • The "task" of the $T$-flow is not to include the coupling to reservoirs but instead to reduce its effect from the value at $T=\infty$ to the value at finite $T < \infty$. At no point is the coupling switched off such that the notion of "(in)elastic" becomes a meaningful way of characterizing the approximations as the referee suggests. One cannot talk about the "energy transfers" that the referee has in mind since these presuppose one can distinguish the energy of the dot from the energy of the reservoirs (which is unavailable) and that one can ignore the energy contribution $\Gamma$ of the coupling (which is fully accounted for at $T=\infty$).

(c) Most importantly, in our approach there is also no need to consider the suggested concepts:

  • As our technical development shows, in order to define, derive and implement the $T$-flow there is no need for the standard perturbation expansion or any terminology that presupposes it.

  • As our intuitive discussion emphasizes, the simple insight of Wilson is sufficient to derive the required equations and suggests the explored approximation scheme. The only point of systematic improvement is how far one goes in the expansion in the renormalized vertices of the $T$-flow equations, which is essentially controlled by the strength of the memory effects, i.e., how low in temperature (!) one goes.

  • As our results show, the approach is surprisingly successful underscoring the importance of Wilson's physical insights about correlations.

In summary, it is a key feature (!) of the $T$-flow method that commonly used concepts and simple pictures -referred to by the referee- are not applicable. It is this completely opposite approach [see point (b)] that reveals a natural connection to Wilson's insightful physical considerations.

With the above in mind, we stress that our calculations do not have a mere "formal" character as suggested by the referee: They are guided down to the details by the physical considerations of Section 2. As mentioned there the renormalized perturbation theory gives a direct handle on the relevant time-correlations which are crucial to the low-temperature (!) physics. The ordinary perturbation expansion --and its associated considerations-- do not reveal the relevant correlations since the $T=\infty$ and $T < \infty$ contributions are completely mixed up. Our paper points out -where possible and applicable- the qualitative underpinnings appropriate to the actually presented technical development. That these are not the "traditional" concepts used in RG methods starting from the bare perturbative expansion underscores the novelty of the presented method. That the equations implied by Wilson's considerations do not have a "simple" rationalization underscores the nontrivial nature of the problem. Clearly, further understanding along these lines is desirable but seems beyond the scope of our first presentation of this novel method.

Revision We fully understand the referee's motivation for the above requests and have highlighted in the Summary and outlook section 6 of the revised submission why "traditional" considerations fail to apply.

(B) Renormalization of the kinetic equation

The latter [virtual processes (energy transfers larger than T)] usually yield renormalization of the quantities entering the effective kinetic equation.

The renormalization of the kinetic equation by higher order contributions which the referee inquires about can be seen in several instances in the paper. The simplest example is the renormalized Liouvillian Eq. (15) which accounts for all orders of tunneling $\Gamma$ at $T=\infty$. Further renormalizations of the kinetic equation are generated by subsequent $T$-flow steps thereby accounting for the finite-temperature dependence of the kinetic equation. Thus, the $T$-flow can be considered as a continuous renormalization of the kinetic equation going from one temperature $T$ to the next lower temperature $T-\delta T$.

Revision This is now pointed out after Eq. (16) and (19) in the revised manuscript (omitting the distinction between elastic and inelastic processes mentioned by the referee which is not applicable here).

(C) Initial state dependence

Further, it is not quite clear whether the procedure depends on the initial density matrix of the system

Revision This is now mentioned more explicitly on p. 5 after Eq. (6) of the revised manuscript.

The existence of a well-defined propagator independent of the initial system state is guaranteed from the very beginning by our assumption of the initial decoupling of system and environment. Related to this is a non-trivial physical property that the exact superoperator $\Pi(t)$ obeys, called complete positivity. It asserts that the evolution correctly treats any entanglement that the system may have with other systems. We have now verified that the approximate propagator obtained by the $T$-flow indeed obey this crucial property for all times by numerical analysis.

Revision In the beginning of Section 5 the nontrivial complete positivity check on our propagator results is now reported and its physical significance is briefly mentioned with pertinent references.

(D) Decay to stationary state

[it is not quite clear] how equilibration processes that lead to thermalization of the system state are described in this approach.

We are not sure what the referee aims at here and similarly under point 3:

... a situation with different temperatures of the reservoirs, which would correspond to a non-equilibrium steady state of the system.

The phrasing of these question seems to suggests that our results deal with transient approach to equilibrium which is not the case: due to the finite bias voltage we already have a non-equilibrium stationary quantum dot state. (Also here we suspect the confusion is due to the picture of 1-dim. wire versus 0-dim. quantum dot, see concluding remark.)

Revision This important point is now highlighted at the beginning of Section 4.

To avoid any confusion we note that:

  • The decay of the initial state of the quantum dot is completely dictated by the memory kernel $K$. This decay is not "thermalization" since it leads to non-equilibrium stationary state.

  • The $T$-flow method does not rely on any assumption about the stationary state, e.g., whether it is unique.

  • The presented results benchmarked the $T$-flow method under strong non-equilibrium conditions for the example of strongly interacting quantum dot at finite bias voltage exceeding both $T$ and $\Gamma$.

  • Introducing (in addition to the voltage bias $\mu_L - \mu_R$ already studied) a finite temperature bias $T_L-T_R$ is possible as mentioned in the Outlook Section 6. This would drive the system out of equilibrium further/differently.

$3$. Present the flow equations for the general case of non-equal temperatures of the reservoirs.

The method is initially formulated for the general case of distinct temperatures of reservoirs; however, all the results are given for the case of equal temperatures. It is extremely interesting to see how the method is applied to a situation with different temperatures of the reservoirs,...

But even without performing the numerical analysis of the flow equations in this intriguing case, it would be nice to present the set of such equations explicitly.

We are glad the referee finds the ability to deal with different temperatures interesting.

Revision In the revised manuscript we have added a new Appendix D, which discusses the flow equations for different temperatures. Additionally, in the Outlook Section 6 we have further indicated which applications we envisage to thermoelectric transport through correlated system by providing two relevant references.

Remark

From the suggestions made by the referee in point 1

.. for example, a derivation of the Luttinger-liquid renormalization of the finite-temperature conductance through a barrier (either in the limit of a weak barrier, or in the limit of weak interaction) in a quantum wire would be very instructive.

and in point 3

In particular, whether the notion of "non-equilibrium dephasing" (see, e.g., papers by Gutman et al. on non-equilibrium bosonization) would naturally emerge in this setting.

we infer a strong affinity with transport through strongly interacting wires with impurities. This is a much more complicated problem than that addressed in our paper and the methods and "pictures" appropriate to that field are quite different from the ones discussed in the present paper. (E.g. the notion of "energy" in point 2.)

In our case we focus on the impurity, a strongly interacting 0 dimensional object (quantum dot) coupled to reservoirs treated as non-interacting. This is clearly not appropriate for 1-dimensional wires where the "reservoir interactions" are well-known to be important (Luttinger-liquid physics). Due to our focus on a relatively simpler situation we are able to address the more complicated issue of transient phenomena.

Revision To avoid confusion we now emphasize from the outset the importance of the assumption of non-interacting reservoirs in the revised manuscript in Section 2.

Attachment:

paper-colored-changes.pdf

---

## Round 1 · Referee Report · Anonymous (Referee 2) · 2022-2-23

Strengths

1) interesting new method with high potential 2) thoroughly tested in a simple model 3) finds re-entrant effect in transient dynamics

Weaknesses

1) sometimes a little guesswork is needed to find the definitions of concepts used in the paper 2) paper uses american spelling (I suppose this is an european journal and would have preferred to see european spelling)

Report

This is an interesting work on the temperature renormalisation scheme, starting from high temperatures and slowly reducing T until one reaches the low temperatures of physical interest. The formalism is well explained and is built around the central self-consistency equation (32).
The working of the method is explored in a single-impurity Anderson model, first for stationary solutions (where results are compared with the one of other general approaches, mainly numerical and are found to agree well) and then also for transient behaviour. An amusing re-entrant behaviour of occupation number and correlations is found and is related to the non-semi-group nature of the dynamics.

An intrinsic assumption of this approach is that the progressive lowering of the temperature can also be slow enough as not to interfere with any intrinsic time-scales of the system. I suspect that feature should limit the applicability of the technique to systems without phase-transition phenomena at some critical temperature T_c. Of course the single-impurity Anderson model studied does not have such a phase transition.

But this remark is not meant to cast into doubt the general interest of the method proposed in this work.

Requested changes

please give a more detailed definition of the `bias' you use on p. 10 -- in the present version one has to guess what you probably mean
(you seem to consider two baths, called right and left and use two parameters \mu_L, \mu_R to finally define your bias V).

  • validity: high
  • significance: high
  • originality: high
  • clarity: high
  • formatting: excellent
  • grammar: excellent

Author:  Konstantin Nestmann  on 2022-03-17  [id 2293]

(in reply to Report 2 on 2022-02-23)
Category:
answer to question

We thank the referee for the report and are glad he found our manuscript interesting and well explained.

**Revision** As suggested we now define the bias $V=\mu_L - \mu_R$ on p. 10 of the manuscript.

**Revision** We have checked spelling for British English.

**Revision** In further extending our numerical results in response to a question by referee 1 we noted that Fig. 2b contained curves which were insufficiently converged. This has been corrected without affecting any part of the text and conclusions.

The suggested question regarding the applicability of our method to systems with phase transitions is intriguing, but beyond the scope of the present manuscript.

Attachment:

paper-colored-changes_J4mBAnW.pdf

---

## Round 2 · Referee Report · Anonymous (Referee 2) · 2022-3-19

Report

all requested changes were taken into account in a satisfactory way.
I recommend to accept for publication.

---

## Round 2 · Referee Report · Anonymous (Referee 1) · 2022-3-21

Strengths

same as in my first report

Weaknesses

-

Report

I tend to accept the authors' arguments regarding my comments from the first report. I also appreciate the changes made to the manuscript in response to my questions. The article can now be published as it is.

Requested changes

none

---

## Round 2 · List of Changes

• Added a new Appendix D, which discusses the flow equations for different temperatures
  • Added the new Appendix F "Exact solution for $U=0$"
  • Several clarifications in response to the referees (see replies)
  • We noted that Fig. 2b contained curves which were insufficiently converged. This has been corrected without affecting any part of the text and conclusions.

---

## Editorial Decision

published